# Development of a Cardiopulmonary Exercise Test Protocol Using Aquatic Treadmill in Healthy Adults: A Pilot Study

**DOI:** 10.3390/healthcare10081522

**Published:** 2022-08-12

**Authors:** Hee-Eun Choi, Chul Kim, Hwan-Kwon Do, Hoo-Seok Lee, Eun-Ho Min

**Affiliations:** 1Department of Physical Medicine and Rehabilitation, Haeundae Paik Hospital, Inje University College of Medicine, 875 Haeun-daero, Haeundae-gu, Busan 48108, Korea; 2Department of Physical Medicine and Rehabilitation, Sanggye Paik Hospital, Inje University College of Medicine, 1342, Dongil-ro, Nowon-gu, Seoul 01757, Korea; 3Department of Biomedical Engineering, Kyung Hee University, 1732, Deogyeong-daero, Giheung-gu, Yongin 17104, Korea

**Keywords:** cardiopulmonary exercise test, aquatic treadmill, metabolic equivalent

## Abstract

Traditional cardiopulmonary exercise test (CPET) protocols are difficult to apply to patients who have difficulty walking on a treadmill. Therefore, this study aimed to develop an aquatic treadmill (AT) CPET protocol involving constant increments in exercise load (metabolic equivalents (METs)) at regular intervals. Fourteen healthy male participants were enrolled in this study. The depth of the water pool was set to the umbilicus level of each participant, and the water temperature was maintained at 28–29 °C. The testing protocol comprised a total of 12 stages at different speeds. The starting speed was 0.7 km/h, which was increased by 0.6 or 0.7 km/h every 2 min. Heart rate, blood pressure, oxygen uptake, minute ventilation, respiratory exchange ratio, and rate of perceived exertion were recorded at each stage. All values showed a significant increasing trend with stage progression (*p* < 0.001). Peak oxygen uptake and heart rate values were 29.76 ± 3.75 and 168.36 ± 13.12, respectively. We developed a new AT CPET protocol that brings about constant increments in METs at regular intervals. This new AT CPET protocol could be a promising alternative to traditional CPET protocols for patients who experience difficulty walking on a treadmill.

## 1. Introduction

The cardiopulmonary exercise test (CPET) is a key component of the initial assessment for cardiac rehabilitation programs [1]. CPET provides a comprehensive assessment of cardiovascular, respiratory, and metabolic responses to exercise. The treadmill is the most commonly used dynamic exercise testing device. Several validated treadmill testing protocols exist, including the Bruce, Modified Bruce, Naughton, and Balke protocols, each comprising different exercise durations, speeds, and grades [2,3,4]. However, it is difficult to perform treadmill testing in people with obesity, gait problems, balance impairments, lower extremity pain, or other issues related to deconditioning.

Several bicycle ergometer testing protocols have been developed as alternatives to treadmill tests. However, a major limitation of cycle ergometer testing is fatigue of the quadriceps muscles that can limit test tolerance. Leg fatigue in an inexperienced subject could cause early test termination before reaching a true VO_2max_ [1]. Thus, VO_2max_ is reported to be 10–20% lower in cycle versus treadmill testing [5,6,7,8].

The aquatic treadmill (AT) is an underwater exercise program with weight supporting effects according to buoyancy. Immersion to the symphysis pubis, umbilicus, and xiphoid process reduces weight bearing by 40%, 50%, and 60%, respectively [9]. This technique is beneficial for many populations, including the elderly, arthritic, injured, and obese populations [10]. Furthermore, the viscosity and cushioning effect of the water allow people with balance or gait problems to walk confidently [11]. There are two types of aquatic running: deep-water running and shallow-water running. In the deep-water method, runners are supported by a buoyant vest or belt and suspended so their feet do not touch the floor. Therefore, deep-water running has quite different characteristics from land running because of the absence of the ground reaction phase. In the shallow-water method, runners are typically only submerged up to the waist or chest and can run in a similar fashion to land running.

Several prior studies have compared the cardiorespiratory response of AT exercise with land treadmill (LT) exercise. Some studies have demonstrated that AT exercise produces a lower cardiorespiratory response than LT exercise [12,13,14], while other studies have reported similar cardiorespiratory responses for both [15,16,17]. In contrast, Gleim and Nicholas [18] demonstrated a higher cardiorespiratory response during AT exercise than during LT exercise. These conflicting results can be attributed to different experimental settings such as water level and treadmill speed.

Electrocardiogram (ECG) provides meaningful information regarding heart performance and functionality during exercise. However, ECG signal quality is dramatically decreased in water-submerged conditions because conventional Ag/AgCl electrodes show poor performance when penetrated with water. To correct this limitation, new ECG electrodes made of graphite pencil lead have been developed [19]. Compared with conventional Ag/AgCl electrodes, these graphite electrodes provide better ECG waveform quality in underwater conditions. Therefore, they may be more suitable for research involving underwater ECG monitoring.

Thus, we reasonably hypothesized that AT could be used to perform CPET in patients who experience difficulty walking on a LT. We further hypothesized that an incremental increase in exercise load could be achieved in such an approach by gradually increasing AT speed. To date, a CPET protocol using AT has not yet been established. Therefore, the purpose of this study was to develop a new, standardized AT CPET protocol based on these hypotheses. 

## 2. Materials and Methods

### 2.1. Participants

Participants were recruited through poster advertisements placed in the outpatient clinic and lobby of Inje University Haeundae Hospital. Fourteen healthy male participants were recruited from May 2019 to June 2019. The inclusion criteria were as follows: (1) healthy adults in their 20s and 30s, (2) absence of underlying medical conditions, (3) ability to perform aquatic exercise, and (4) voluntary, informed participation in the study. The exclusion criteria were as follows: (1) individuals with aquaphobia, (2) inability to enter water due to skin issues, (3) refusal to provide informed consent, and (4) other contraindications to exercise testing as identified by the American College of Sports Medicine [20].

The study was conducted according to the guidelines of the Declaration of Helsinki and was approved by the Inje University Haeundae Hospital Institutional Review Board (approval no. 2018-10-006). All participants provided written informed consent after receiving a detailed explanation of the study protocol. In addition, this study was registered at the Clinical Research Information Service (approval no. KCT0004002).

### 2.2. CPET with AT

The participants were instructed not to consume food or caffeine drinks for 4 h prior to the test. They received a detailed explanation of the procedure and purpose of the test, including symptom and sign end points, and possible complications. We measured resting heart rate (HR) and blood pressure (BP) in the sitting position. A face mask for respiratory gas analysis was worn tightly to prevent air leakage. For underwater ECG monitoring, ten graphite electrodes were attached to the participants’ chest (Figure 1 and Figure 2) [1,19,21]. While the electrodes are not submerged in water, vigorous exercise can cause water to splash onto the electrodes. Therefore, we used waterproof electrodes to overcome the poor ECG signal quality associated with conventional electrodes when penetrated by water.

The participants entered the water tank and were immersed in water up to their umbilicus. The water and room temperatures were maintained at 28−29 °C and 25−26 °C, respectively. The face mask was connected to a respiratory gas analyzer (Quark CPET, COSMED, Rome, Italy). The graphite electrodes were connected to 12 channel ECGs for real-time CPET (CASE, GE healthcare, Chicago, IL, USA). The test was performed on Aquatrac-2000 (NARAMED, Gwangju, Korea) (Figure 3).

### 2.3. Testing Protocol

The participants stood at rest in the water for 2 min before the test. The AT CPET protocol comprised 12 stages at different speeds with no incline. The starting speed was 0.7 km/h (stage 1) and was increased by 0.6 or 0.7 km/h every 2 min. HR, BP, oxygen uptake (VO_2_/kg), metabolic equivalents (METs), minute ventilation (VE), and respiratory exchange ratio (RER) were recorded at each stage. The rate of perceived exertion (RPE) was assessed using the 6-to-20 Borg scale at each stage [1]. The measurement timing of each outcome was set according to the American College of Sports Medicine guidelines [20].

Termination of the CPET was determined according to the American College of Sports Medicine guidelines: (1) RPE of 17 (hard to very hard), (2) RER of >1.10, (3) reaching age-adjusted maximum HR (220-age), (4) ischemic ECG changes, (5) severe arrhythmias such as ventricular tachycardia, (6) signs of poor perfusion such as cyanosis or sudden pallor, (7) participant’s request to stop, and (8) other emergencies as judged by the research staff [1,20]. After the test, participants stood in the water for 2 min to cool down. 

### 2.4. Statistical Analysis

The recorded data of all participants were averaged for each test stage and expressed as the mean and standard deviations. The repeated measure ANOVA was used to evaluate the trend of each outcome according to stage progression. All statistical analyses were performed with IBM SPSS software version 25.0 (IBM Corporation, Armonk, NY, USA). Statistical significance was set at *p* < 0.05.

## 3. Results

### 3.1. Demographics and Baseline Parameters

Participants’ demographics and baseline parameters are shown in Table 1. The mean age and BMI of participants were 30.0 ± 4.9 years and 24.8 ± 2.2 kg/m², respectively. The mean resting HR was 79.7 ± 9.4 beats/min, and the mean resting systolic BP (SBP) and diastolic BP (DBP) were 125.5 ± 17.0 mmHg and 77.9 ± 9.9 mmHg, respectively. The mean immersion HR was 73.9 ± 7.4 beats/min, and the mean immersion SBP and DBP were 126.4 ± 18.8 mmHg and 77.4 ± 7.3 mmHg, respectively.

### 3.2. The Outcomes of CPET Using AT for Each Stage

The CPET outcomes for each stage are shown in Table 2. Since seven participants could not complete stage 12, only data for stages 1–11 were included. HR, VO_2_/kg, METs, VE, RER, and RPE values showed a significant increasing trend with stage progression (*p* < 0.001). Peak VO_2_/kg and HR values were 29.76 ± 3.75 mL/min/kg and 168.36 ± 13.12 beats/min, respectively. The mean RER value exceeded 1.0 by stage 9 but had not reached 1.1 at stage 11. The mean RPE value exceeded 15 at stage 8, and a maximal mean value of 18.00 ± 1.18 was reached at stage 11. 

### 3.3. Cardiorespiratory Responses during CPET Using AT

Figure 4 shows the cardiorespiratory responses during CPET using AT. HR, VO_2_/kg, METs, VE, and SBP values all showed an increasing trend with stage progression, while DBP showed no such trend. 

### 3.4. Development of the AT CPET Protocol

In our testing protocol, approximately 2, 3, 4, 5, 6, 7, and 8 METs were measured at stages 2, 4, 6, 7, 8, 9, and 10, respectively. We proposed a new AT CPET protocol that allows a constant increment in METs at regular intervals. Table 3 shows the comparison of data from our new protocol with data from the modified Bruce protocol.

## 4. Discussion

The results of the present study demonstrated a statistically significant increasing trend in cardiorespiratory outcomes and RPE with stage progression during AT CPET. This study also further confirmed the validity of the graphite electrodes for underwater testing, which worked well and showed good ECG waveform quality. Furthermore, we developed a novel AT CPET protocol that can bring about constant increments in METs at regular intervals. Since our protocol showed an increase of 1 MET for every stage, it can provide more detailed information regarding the exercise capacity of patients who cannot perform CPETs on a LT. To the best of our knowledge, this is the first study aimed at developing a CPET protocol using AT. This CPET protocol may be applied to diverse patient groups with difficulty walking on the treadmill.

In this study, the average resting HR after immersion was 5.8 beats/min lower than before immersion, while the average maximal HR was 22 beats/min lower than the age-predicted maximum. This significantly lower HR in water is thought to be due to the effect of hydrostatic pressure on the body, which causes an increase in venous return and stroke volume, allowing for the maintenance of cardiac output with a lower HR [22]. The cardiovascular reflex response to cold receptors may also contribute to the lowering of HR in water, as the water temperature of 28–29 °C in our study was lower than the thermoneutral temperature of 35 °C [18]. Several previous studies have also reported a decrease in HR during underwater exercise compared to land exercise [12,23,24,25].

The peak VO_2_ measured in our testing protocol (29.76 ± 3.75 mL/min/kg) was significantly lower than that measured in the Bruce protocol (42.0 ± 5.0 mL/min/kg) when performed by healthy Korean men in their 30s [26]. This result is consistent with several other previous studies reporting lower peak VO_2_ values during underwater exercise compared with land exercise [12,13,14,27]. Dowzer et al. [12] reported that underwater running in waist-level water produced a 16% lower peak VO_2_ than running on LT. Town and Bradley [13] also observed a 10% reduction in peak VO_2_ during shallow-water running compared with land exercise. This decrease in peak VO_2_ values during underwater exercise has been attributed to reduced overall workload due to the buoyancy effect and reduced limb load, thus leading to reduced cardiorespiratory response [17,28]. It can also be explained by increased anaerobic metabolism during underwater exercise as a result of reduced lower limb perfusion due to hydrostatic pressure [12]. Additional recruitment of smaller muscle groups during underwater exercise can also contribute to the greater involvement of anaerobic metabolism [29]. Anaerobic metabolism can cause leg fatigue and pain, which may prevent participants from reaching maximal aerobic exercise capacity. A study by Astorino et al. [30] revealed that long duration CPET protocols produced significantly lower VO_2_ max values than short or middle duration protocols [30]. Our testing protocol required more time than other CPET protocols to reach the same exercise intensity. For example, in our testing protocol, it took 18 min to reach 7 METs, whereas in the modified Bruce protocol, it took 12 min. It is thought that a longer test duration leads to increased body temperature, resulting in peripheral vasodilation and dispersion of accumulated heat, which in turn leads to decreased stroke volume and maximal cardiac output [31]. Therefore, we picked seven stages that can bring about constant increments in METs at regular intervals and developed a new protocol that would take a total of 14 min.

Resting BP did not change significantly after water immersion. This was consistent with previous studies, which reported no changes in arterial pressures with water immersion [22,32]. During exercise, SBP showed a tendency to increase with stage progression, but DBP remained constant. It is known that SBP rises with increasing workload as a result of increasing cardiac output, while DBP usually remains the same or is decreased because of vasodilatation of the vascular bed [1]. 

Water depth has a profound effect on the magnitude of buoyancy. Immersion to the symphysis pubis, umbilicus, and xiphoid process reduces limb loading by 40%, 50%, and 60%, respectively [9]. This means that water-submersion level has a significant influence on cardiorespiratory responses during underwater exercise. Gleim and Nicholas [18] demonstrated that underwater treadmill exercise in water levels up to the ankle, patella, and midthigh produced higher VO_2_ and HR values compared with land running. They also reported that running in waist-deep water yielded VO_2_ values comparable to those seen during LT running. Napoletan and Hicks [33] reported that running in chest-deep water produced significantly lower VO_2_ than running in thigh-deep water. When walking underwater, the ground reaction force has been shown to decrease with increasing water depth [34]. The water depth in our study (to the umbilicus) was chosen to achieve VO_2_ costs as close as possible to those of LT walking at similar speeds. Considering the potential application of this protocol to diverse patient population groups, we tried to set the water depth to a level that would not cause dyspnea while allowing normal gait without floating during stride cycle. 

The effect of water temperature on cardiorespiratory response is also an important factor to consider. A water temperature of 35 °C is considered thermoneutral for resting water immersion, and 29–33 °C is considered thermoneutral during dynamic exercises [35]. Water immersion in temperatures below thermoneutral causes peripheral vasoconstriction, resulting in an increase of the central blood volume and decrease of HR. Craig and Dvorak [36] also suggested that water temperatures over 30 °C could elicit similar HR responses to those seen out of water, whereas a temperature below 30 °C might elicit a lower HR [36]. When exercising above thermoneutral temperatures, HR increases rapidly due to a greater cardiac output and blood flow to the skin [37]. As the water temperature was set at 28–29 °C (below thermoneutral) in our study, the average resting and peak HR values were comparatively lower than those of the land environment. 

A prior study compared the performances of graphite electrodes with Ag/AgCl electrodes, and further analyzed the electrode-to-skin contact impedance [19]. Graphite electrode impedance was smaller than Ag/AgCl electrode impedance. Performance was analyzed and compared under various conditions: dry surface, freshwater immersion with/without movement, freshwater wet, saltwater immersion with/without movement, and saltwater wet conditions. This study showed that the graphite electrodes provided all morphological components of ECG signals, in all conditions. Especially in freshwater and saltwater immersion, graphite electrodes provided better ECG signal quality than Ag/AgCl electrodes. These graphite electrodes are inexpensive but have high efficiency. 

### Limitations and Future Perspectives

This study has some limitations. First, this protocol can be applied only under similar environmental conditions, including similar air and water temperatures. Second, since the participants did not perform standardized CPET such as the modified Bruce protocol, we did not have baseline VO_2_ max data to which our results could be compared. Third, since the participants were all males in their 20s and 30s, the results are not generalizable to all adults. It is presumed that no women volunteered because of the need to attach ECG electrodes to the chest surface and enter into the water. Fourth, this study enrolled a relatively small number of participants, which may have reduced the statistical power of the study and increased margin of error. We look forward to conducting future studies that apply our newly developed protocol to a larger and more diverse population. With further validation, this AT CPET protocol could be widely used to measure exercise capacity and cardiorespiratory response in patients who cannot perform conventional CPETs.

## 5. Conclusions

This pilot study developed a novel AT CPET protocol that brings about constant increments in METs at regular intervals. We also confirmed the validity of the graphite-based electrodes for underwater ECG monitoring, through their effective use in this study. Considering the weight supporting effect of water, this new AT CPET protocol could be a promising alternative to traditional CPET protocols for patients who have difficulty walking or running on regular treadmills.

## Figures and Tables

**Figure 1 healthcare-10-01522-f001:**
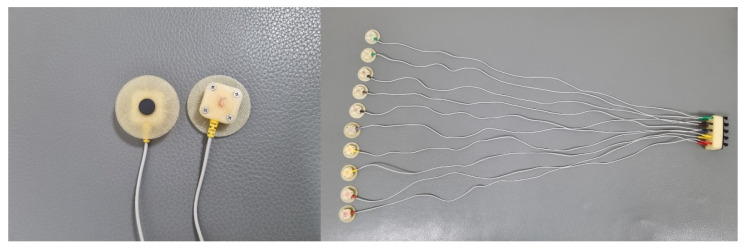
Graphite-based electrocardiogram electrodes.

**Figure 2 healthcare-10-01522-f002:**
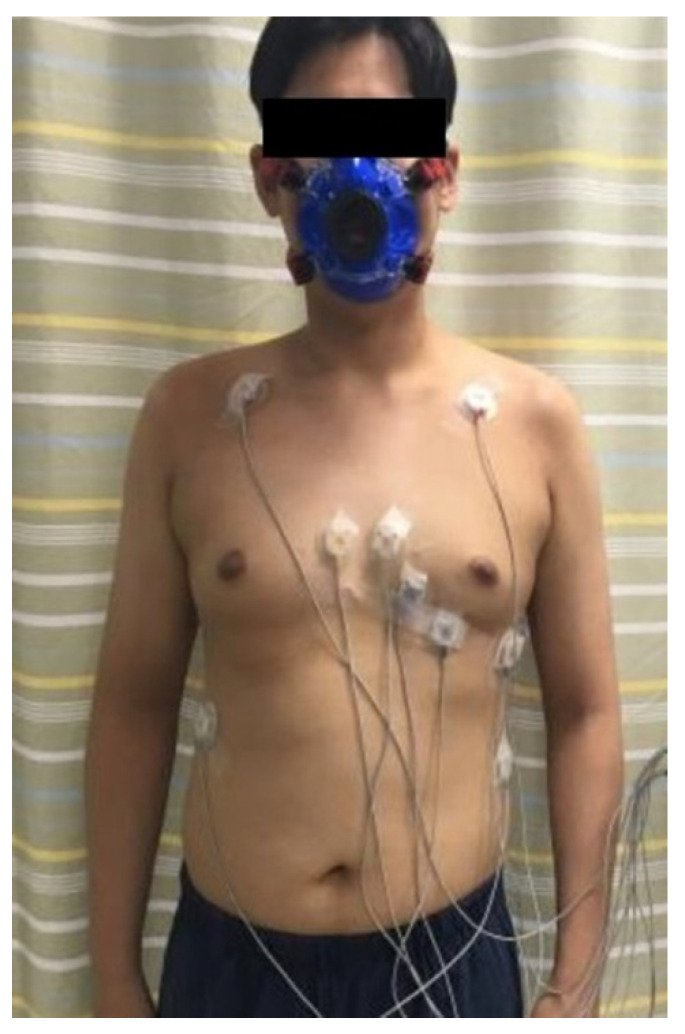
Location of electrocardiogram electrodes.

**Figure 3 healthcare-10-01522-f003:**
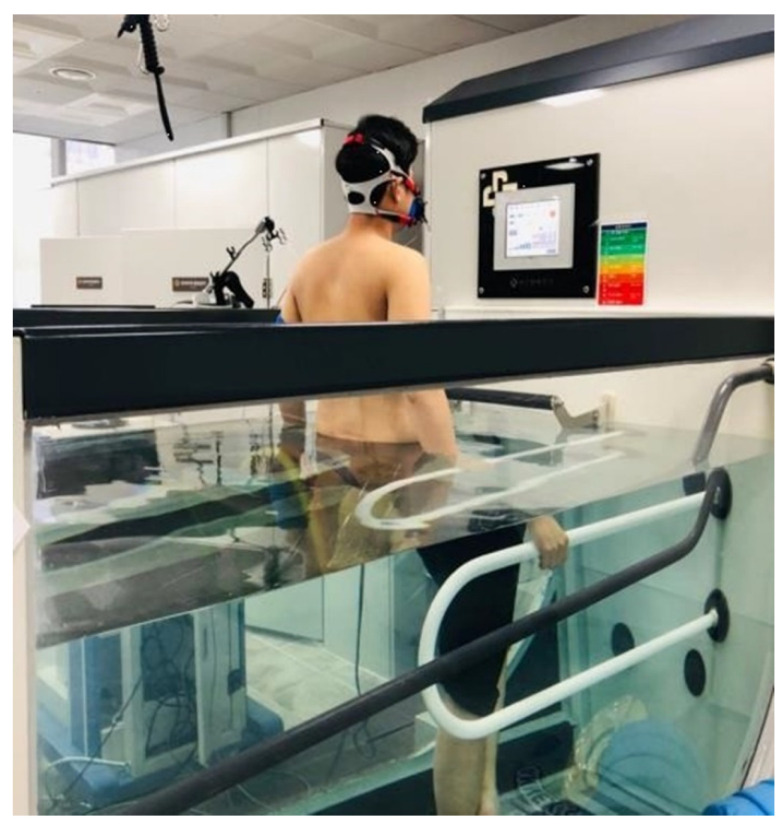
Aquatic treadmill.

**Figure 4 healthcare-10-01522-f004:**
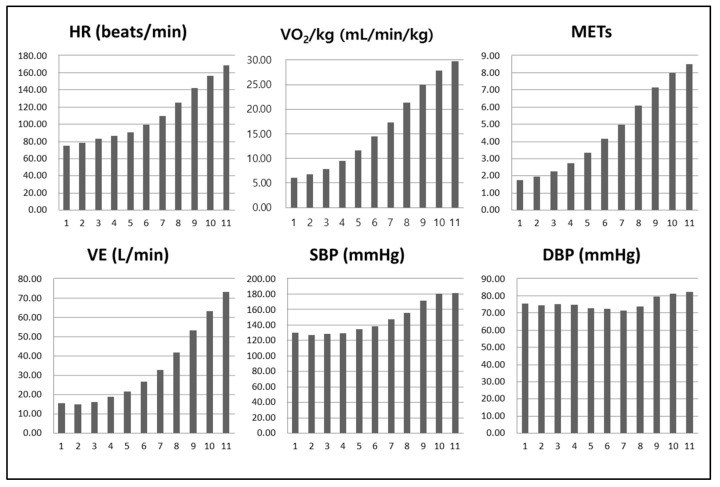
Cardiorespiratory responses during cardiopulmonary exercise test. HR, heart rate; VO_2_, oxygen uptake; METs, metabolic equivalents; VE, minute ventilation; SBP, systolic blood pressure; DBP, diastolic blood pressure.

**Table 1 healthcare-10-01522-t001:** Participants’ demographics and baseline parameters.

Variables	Values
Age (years)	30.0 ± 4.9
Height (cm)	175.6 ± 4.2
Body weight (kg)	76.4 ± 8.1
BMI (kg/m²)	24.8 ± 2.2
HR_rest_ before immersion (beats/min)	79.7 ± 9.4
SBP_rest_ before immersion (mmHg)	125.5 ± 17.0
DBP_rest_ before immersion (mmHg)	77.9 ± 9.9
HR_rest_ after immersion (beats/min)	73.9 ± 7.4
SBP_rest_ after immersion (mmHg)	126.4 ± 18.8
DBP_rest_ after immersion (mmHg)	77.4 ± 7.3

BMI, body mass index; HR, heart rate; SBP, systolic blood pressure; DBP, diastolic blood pressure. Data are presented as mean ± SD.

**Table 2 healthcare-10-01522-t002:** CPET outcomes.

Stage	Speed (km/h)	HR (Beats/min)	VO_2_/kg (mL/min/kg)	METs	VE (L/min)	RER	RPE
1	0.7	74.64 ± 9.25	6.06 ± 0.98	1.73 ± 0.28	15.39 ± 3.49	0.90 ± 0.10	6.93 ± 1.44
2	1.4	78.14 ± 10.48	6.80 ± 1.03	1.94 ± 0.29	14.82 ± 2.85	0.83 ± 0.08	7.71 ± 1.27
3	2.0	82.93 ± 9.50	7.85 ± 1.12	2.24 ± 0.32	16.00 ± 2.58	0.80 ± 0.06	8.79 ± 1.42
4	2.7	86.29 ± 9.18	9.50 ± 1.21	2.71 ± 0.35	18.67 ± 3.51	0.80 ± 0.06	10.29 ± 1.54
5	3.3	90.50 ± 9.65	11.70 ± 2.18	3.34 ± 0.62	21.59 ± 3.16	0.79 ± 0.07	11.43 ± 1.60
6	4.0	99.14 ± 10.07	14.55 ± 3.11	4.16 ± 0.89	26.56 ± 3.59	0.83 ± 0.05	12.64 ± 1.91
7	4.6	109.79 ± 11.43	17.38 ± 3.02	4.97 ± 0.86	32.76 ± 4.36	0.89 ± 0.05	13.93 ± 1.82
8	5.3	125.21 ± 14.80	21.34 ± 2.82	6.10 ± 0.81	41.64 ± 4.73	0.94 ± 0.06	15.43 ± 1.09
9	5.9	142.14 ± 15.97	25.05 ± 3.73	7.16 ± 1.07	53.16 ± 8.18	1.01 ± 0.06	16.71 ± 1.07
10	6.6	156.50 ± 17.76	27.94 ± 3.44	7.98 ± 0.98	63.11 ± 10.66	1.04 ± 0.07	17.57 ± 1.09
11	7.2	168.36 ± 13.12	29.76 ± 3.75	8.50 ± 1.07	73.15 ± 9.11	1.07 ± 0.06	18.00 ± 1.18
*p* Value for trend		<0.001	<0.001	<0.001	<0.001	<0.001	<0.001

CPET, cardiopulmonary exercise test; HR, heart rate; VO_2_, oxygen uptake; METs, metabolic equivalents; VE, minute ventilation; RER, respiratory exchange ratio; RPE, rate of perceived exertion. Data are presented as mean ± SD. A *p* value < 0.05 was considered significant.

**Table 3 healthcare-10-01522-t003:** Comparison of our new aquatic treadmill protocol data and modified Bruce protocol data.

New Aquatic Treadmill Protocol					Modified Bruce Protocol				
Stage	Cumulative Time (min)	Speed (km/h)	Grade (%)	METs	Stage	Cumulative Time (min)	Speed (km/h)	Grade (%)	METs
1	1	1.4	0	1.94	1	1	2.7	0	2.9
	2					2			
2	3	2.7	0	2.71		3			
	4				2	4	2.7	5	3.7
3	5	4.0	0	4.16		5			
	6					6			
4	7	4.6	0	4.97	3	7	2.7	10	5.1
	8					8			
5	9	5.3	0	6.10		9			
	10				4	10	4.0	12	7.1
6	11	5.9	0	7.16		11			
	12					12			
7	13	6.6	0	7.98	5	13	5.4	14	10.3
	14					14			

METs, metabolic equivalents.

## Data Availability

The study data are available upon request from the corresponding author.

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
