# Peer review of "Development of a Cardiopulmonary Exercise Test Protocol Using Aquatic Treadmill in Healthy Adults: A Pilot Study"

_healthcare, 2022, doi:10.3390/healthcare10081522_

Round 1
Reviewer 1 Report
I have read very carefully the work entitled "Development of Cardiopulmonary Exercise Test Protocol Using Aquatic Treadmill in Healthy Adults: A Pilot Study.
The Pilot study is very well structured and organized especially in the materials and methods section.
The results are very clear and highlight all the goals put forward by the authors.
Some points emerge that I think it is important to deepen:
1) Improve Native English
2) General Spelling check
3) Highlight the limitations of the study: the limited number of patients in particular.
Reviewer 2 Report
Dear Authors,
you presented a well-written manuscript describing a cardiopulmonary exercise test protocol based on aquatic treadmill. You provided a comprehensive description of the protocol and pathophysiology elements influencing exercise in water. Such an exercise test may find useful implementation in the clinical praxis, enabling the cardiopulmonary evaluation of patients who are not able to perform a land treadmill test. Please pay attention to the following comments and questions:
Abstract: please explain the abbreviation METs (metabolic equivalents) upon first appearance.
2.1 Subjects: please explain why you chose only male participants for your study.
2.2 CPET with AT: please explain why you had to develop waterproof graphite-based electrocardiogram electrodes for the chest if the water in the tank was maximally reaching the umbilicus lever and therefore, no contact was expected between the water and the electrodes.
With Best Regards
Reviewer 3 Report
Dear authors,
I am pleased to have reviewed your research. It is a good study, however, I think that a lot of things need to be improved for it to be published.
The introduction is too brief and goes into too much superfluous depth about the problems addressed. I think it should be taken further. I would also like to name and comment on some research similar to this one.
The research lacks research hypotheses. Please provide them just before the research objective.
Please indicate how the research participants were contacted in order to participate in the research.
The limitations of the research are noted, however, these are at the end of the discussion. Please create a section entitled "Limitations and Future Perspectives" and add the limitations there.
Develop more fully the conclusions of the research.
Revise the bibliography because the journal names have not been cited according to journal guidelines.
Round 2
Reviewer 3 Report
Dear authors.
The research has been significantly improved. I hereby approve it in the present form.
Congratulations on the work done